# Limitations of scRNA-seq Zero-Imputation Methods for Network Inference

**Ankit Bhardwaj** [1]  **Joshua Weiner** [2]  **Preetha Balasubramanian** [3]  **Lakshminarayanan Subramanian** [1]

## Abstract

Zero-imputation methods are widely applied to address non-biological zeros in scRNA-seq data. However, these methods can introduce artificial signals, skewing the results of downstream analysis to match initial assumptions rather than emulate the underlying biological processes. This paper makes a simple but surprising observation: we demonstrate that several popular zero imputation techniques provide significantly varied results on the downstream network inference tasks over the same real-world scRNA datasets. Benchmarking their performance on synthetically controlled simulated scRNA datasets using the SERGIO simulator and the GENIE3 network inference algorithm, we observed poor metrics across the board. A key takeaway from our analysis is both unearthing the unreliability of existing imputation techniques and the inability to define a uniform gold-standard for zero imputation.

## 1. Introduction

Zero-imputation in single-cell RNA sequencing (scRNA-seq) data refers to the process of addressing the presence of zero values in the gene expression matrix. In scRNA-seq data, zeros can arise from several sources. Technical dropouts occur due to the low amounts of mRNA in individual cells, causing some genes to be missed during sequencing, leading to false zeros. Biological zeros represent the genuine absence of gene expression in certain cells, while sampling variability reflects the inherent variations in capturing and sequencing mRNA, resulting in some genes not being detected in some cells (Jiang et al., 2022).

Proper handling of these zeros is essential to maintain data integrity and ensure accurate biological interpretations. Correcting for dropouts can reduce biases and enhance the reliability of downstream analyses, such as clustering, differential expression analysis, and trajectory inference. Several methods exist for zero-imputation including statistical models, Bayesian frameworks, and ML algorithms (MAGIC (van Dijk et al., 2018), SAUCIE (Amodio et al., 2019), sc-Scope (Deng et al., 2019)), etc. A comprehensive review of different zero imputation methods for scRNA-seq data can be found in (Lähnemann et al., 2020).

Some researchers argue against zero-imputation in single-cell RNA sequencing (scRNA-seq) data due to concerns about introducing artificial signals that can lead to incorrect biological interpretations (Svensson, 2020; Kim et al., 2020; Qiu, 2020).

Motivated by this dichotomy of arguments across the two philosophies of thought on zero imputation, we embarked on two simple experiments in this paper to determine if we can define a gold standard for measuring the quality of zero imputation methods. In the first experiment, we consider real-world scRNA datasets combined with a standard set of zero imputation methods and use the GENIE3 gene regulatory inference algorithm (Huynh-Thu et al., 2010) to examine differences in the inferred gene regulatory networks. We observed significant variability in the results with a low Jaccard similarity measure between the inferred gene regulatory networks across these datasets.

In the second experiment, we leveraged a state of the art simulation framework, SERGIO (Dibaeinia & Sinha, 2020) where we can pre-define the ground truth gene regulatory network to generate synthetic scRNA data that resembles real-world experimental scRNA data including modeling the noise generated in the biological experimental process. Here, we benchmarked several zero imputation techniques against the ground truth and surprisingly found that none of the techniques were able to satisfactorily recover the ground truth. In fact, we found a phenomenal gap between the ground truth and the inferred network using standard AUCROC metrics.

The key takeaways from both these experiments are two-fold. First, the lack of a clear consensus on best practices for zero-imputation methods leads to significant variability and reproducibility issues in scRNA-seq studies. Second, even defining a ground truth using a synthetic data generation

[1]Department of Computer Science, New York University, NY, USA [2]Palantir Technologies, NY, USA [3]Department of Pediatrics, Weill Cornell Medicine, NY, USA. Correspondence to: Ankit Bhardwaj <bhardwaj.ankit@nyu.edu>, Lakshminarayanan Subramanian <lakshmi@nyu.edu>.

*Accepted at the 1st Machine Learning for Life and Material Sciences Workshop at ICML 2024*. Copyright 2024 by the author(s).

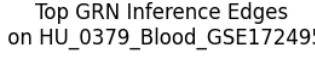

Top GRN Inference Edges
on HU_0379_Blood_GSE172495

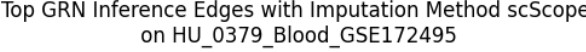

Top GRN Inference Edges with Imputation Method scScope
on HU_0379_Blood_GSE172495

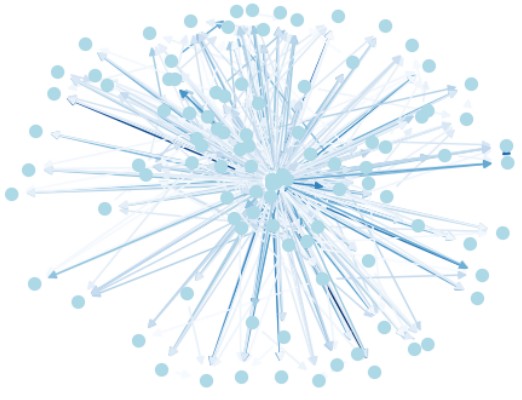

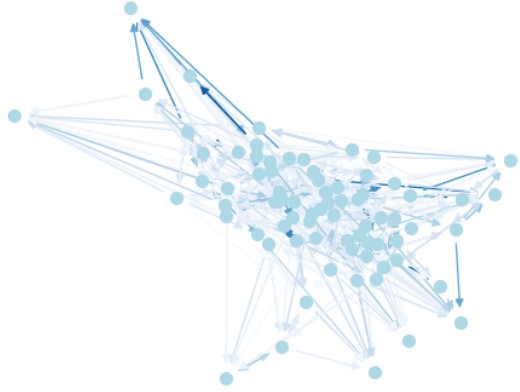

Top GRN Inference Edges with Imputation Method MAGIC
on HU_0379_Blood_GSE172495

Top GRN Inference Edges with Imputation Method SAUCIE
on HU_0379_Blood_GSE172495

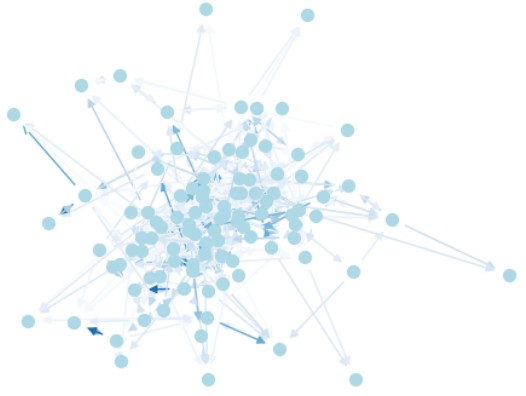

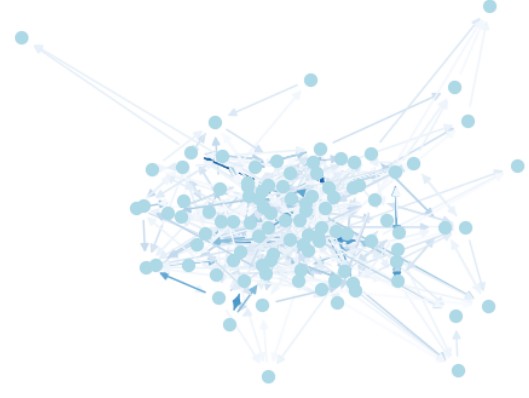

*Figure 1.* Different inferred gene regulatory networks for different imputation pipelines. Limited selection to top 5% of edge weights for gene-pairs for graph clarity.

experiment reveals that existing methods are very far from recovering the ground truth. These simple observations do reveal surprising limitations of zero imputation methods that may even question the validity of results generated using these methods.

## 2. Zero-Imputation Leads to High Variability in Network Inference

Almost all zero-imputation methods typically report results on standard downstream tasks like clustering, heterogeneity analysis, and trajectory inference. However, this approach ignores one important aspect of the scRNA-seq data generation process, that the gene expression in different cells is governed by a network of regulatory relationships between different genes, also called the Gene Regulatory Network

(GRN).

When considering the task of network inference using scRNA-seq data, it is imperative that we understand the effects of different zero-imputation techniques to the final result. For estimating this effect, we took a random subset (number of genes = 10,000) of the scRNA-seq following datasets.

- HU_0379_Blood_GSE172495 (blo)
- HU_0243_Prostate_GSE134355 (pro)
- HU_0325_Esophagus_GSE159929 (eso)
- HU_0231_Pancreas_GSE134355 (pan)
- HU_0269_Uterus_GSE134355 (ute)

We ran the GENIE3 network inference algorithm with different imputation pipelines across these datasets to predict the underlying GRN. We indeed found significant variation in

| | None | SAUCIE | scScope | DeepImpute | MAGIC | SCVI | KNN |
|---|---|---|---|---|---|---|---|
| None | 1.000 | 0.046 | 0.088 | 0.522 | 0.064 | 0.058 | 0.032 |
| SAUCIE | 0.103 | 1.000 | 0.057 | 0.045 | 0.065 | 0.050 | 0.035 |
| scScope | 0.189 | 0.060 | 1.000 | 0.095 | 0.048 | 0.056 | 0.047 |
| DeepImpute | 0.513 | 0.109 | 0.153 | 1.000 | 0.059 | 0.046 | 0.044 |
| MAGIC | 0.008 | 0.026 | 0.016 | 0.013 | 1.000 | 0.059 | 0.034 |
| SCVI | 0.053 | 0.047 | 0.035 | 0.050 | 0.038 | 1.000 | 0.038 |
| KNN | 0.057 | 0.041 | 0.036 | 0.055 | 0.032 | 0.060 | 1.000 |

| | None | SAUCIE | scScope | DeepImpute | MAGIC | SCVI | KNN |
|---|---|---|---|---|---|---|---|
| None | 1.000 | 0.034 | 0.031 | 0.252 | 0.005 | 0.040 | 0.018 |
| SAUCIE | 0.041 | 1.000 | 0.062 | 0.048 | 0.046 | 0.043 | 0.056 |
| scScope | 0.126 | 0.033 | 1.000 | 0.045 | 0.036 | 0.035 | 0.028 |
| DeepImpute | 0.504 | 0.036 | 0.119 | 1.000 | 0.013 | 0.037 | 0.031 |
| MAGIC | 0.004 | 0.029 | 0.009 | 0.000 | 1.000 | 0.040 | 0.066 |
| SCVI | 0.047 | 0.032 | 0.031 | 0.046 | 0.027 | 1.000 | 0.050 |
| KNN | 0.028 | 0.031 | 0.043 | 0.035 | 0.022 | 0.024 | 1.000 |

| | None | SAUCIE | scScope | DeepImpute | MAGIC | SCVI | KNN |
|---|---|---|---|---|---|---|---|
| None | 1.000 | 0.047 | 0.174 | 0.362 | 0.014 | 0.047 | 0.041 |
| SAUCIE | - | 1.000 | 0.042 | 0.059 | 0.030 | 0.037 | 0.025 |
| scScope | - | - | 1.000 | 0.151 | 0.014 | 0.034 | 0.038 |
| DeepImpute | - | - | - | 1.000 | 0.013 | 0.046 | 0.044 |
| MAGIC | - | - | - | - | 1.000 | 0.022 | 0.032 |
| SCVI | - | - | - | - | - | 1.000 | 0.029 |
| KNN | - | - | - | - | - | - | 1.000 |

| Reference | |
|---|---|
| | HU_0379_Blood_GSE172495 |
| | HU_0243_Prostate_GSE134355 |
| | HU_0325_Esophagus_GSE159929 |
| | HU_0231_Pancreas_GSE134355 |
| | HU_0269_Uterus_GSE134355 |

*Figure 2.* Jaccard Similarity between different GRNs after applying different imputation techniques for 5 human datasets.

the final result, which we present using Figure 2, where we show the Jaccard similarity measure between the results after applying different imputation methods as pre-processing steps across the five different datasets. As can be seen from the tables, the similarity between different predicted GRNs is fairly low (less than 0.1 jaccard index in most cases), across all five datasets. A detailed discussion of the imputation methods evaluated can be found in section 4.3. The notable exception are the graphs generated with raw data without imputation and with DeepImpute, where the similarity scores are the highest across all datasets. The direct implication of these results is that the choice of imputation can lead to completely different inferred gene regulatory networks, with very minimal overlap between the edge sets.

For a better visual representation, we show four examples of such networks in Figure 1, derived by running GENIE3 on the HU_0379_Blood_GSE172495 dataset (blo). In the top left, we have the network inferred by taking top 5% edges predicted by GENIE3 (Huynh-Thu et al., 2010) when directly run on the raw dataset without any pre-processing steps. In top right, we applied the scScope imputation along with the set of pre-processing steps described in (Deng et al., 2019) to the data before running the GENIE3 algorithm and taking the top 5% edges. In bottom left, we followed the pipeline from MAGIC method (van Dijk et al., 2018), and in the bottom-right we have followed the pipeline from SAUCIE (Amodio et al., 2019).

As shown, the choice of imputation method can significantly alter the inferred network and particularly the selection of strongest pairs. Unfortunately, there does not seem to be a consensus in the community about the use of such data processing methods. Motivated by this observation, in this paper, we attempt to benchmark the performance of different zero-imputation methods on the downstream task of GRN discovery, with the larger goal to define a gold-standard data processing pipeline for scRNA-seq data.

## 3. Data

### 3.1. Requirement for "Ground Truth"

To benchmark the performance of zero-imputation methods, one common strategy is to work with simulated synthetic data, as the "ground truth" is not available for the real-data generated by sequencing experiments. Some notable scRNA-seq simulators include Splatter (Zappia et al., 2017), scDesign2 (Sun et al., 2021), SymSim (Zhang et al., 2019), powsimR (Vieth et al., 2017), SCSIM (Giguere et al., 2020), etc. One indication of the negligence of benchmarking on the task of network discovery is that none of the above simulators account for GRNs in their data generation process. As such, they are not suitable for benchmarking the performance of zero-imputation techniques on the network inference task. The SERGIO (Dibaeinia & Sinha, 2020) simulator is one of the few that provide us with a mecha-

nism for generating synthetic scRNA-seq data by modeling GRNs, transcriptional dynamics, and technical noise.

SERGIO begins by taking a gene regulatory network as input. In this network nodes represent genes and edges represent regulatory interactions between genes, which can be either activation or repression. The simulator requires two input files, the first: a comma-separated file where each row represents a target gene, the IDs of its regulators, the interaction strengths for each gene pair 'K' (with positive values for activation and negative for repression), and the Hill coefficients for each pair. Master regulators are excluded from this file, but provided in the second, a comma-separated file of master regulators where each row contains the regulator's ID followed by its production rates for each cell type.

### 3.2. Ground Truth Simulation

SERGIO models the dynamics of the concentration of genes using systems of stochastic differential equations that are derived from the chemical Langevin equation (CLE) (Gillespie, 2000). For every cell, the ground-truth mRNA concentration is modeled as

$$\frac{dx_i}{dt} = P_i(t) - \lambda_i x_i(t) + q_i(\sqrt{P_i(t)}\alpha + \sqrt{\lambda_i x_i(t)}\beta)$$

$$P_i(t) = \sum_{j \in R_i} p_{ij}(t) + b_i(t)$$

$$p_{ij}(t) = K_{ij}\frac{x_j(t)^{n_{ij}}}{h_{ij}^{n_{ij}} + x_j(t)^{n_{ij}}}, \text{if j is an activator of i}$$

$$p_{ij}(t) = K_{ij}(1 - \frac{x_j(t)^{n_{ij}}}{h_{ij}^{n_{ij}} + x_j(t)^{n_{ij}}}), \text{if j is a repressor of i}$$

where,

$x_i(t) \rightarrow$ Expression value of gene i

$P_i(t) \rightarrow$ Production rate of gene i

$\lambda_i \rightarrow$ Decay rate of gene i

$q_i \rightarrow$ Noise amplitude in the transcription for gene i

$\alpha, \beta \rightarrow$ Independent Gaussian white noise processes

$R_i \rightarrow$ Regulator set of gene i

$b_i \rightarrow$ Basal production rate of gene i

$p_{ij}(t) \rightarrow$ Regulatory effect of gene j on gene i

$K_{ij} \rightarrow$ Maximum contribution of gene j on gene i

$n_{ij} \rightarrow$ Hill's coefficient

$h_{ij} \rightarrow$ Regulator concentration for half response

The above set of equations is defined for all genes, making the simulator a discrete dynamical system of the system of stochastic differential equations. The details of the simulator can be found in (Dibaeinia & Sinha, 2020), and the implementation is publicly available at the GitHub repository (https://github.com/PayamDiba/SERGIO).

### 3.3. Noise Simulation

The SERGIO simulation environment allows us to simulate three different noise models, corresponding to different aspects of sequencing pipelines (Zappia et al., 2017). These include

*Outlier Genes:* Each gene is marked as an outlier based on a probability set by the user. If a gene is an outlier, its expression levels in all cells are multiplied by a factor drawn from a log-normal distribution; if not, the expression remains unchanged.

*Library Size:* For each cell, a library size parameter is drawn from a log-normal distribution. The expression values of all genes in the cell are then scaled by a constant factor so that the total depth of the cell matches the sampled library size.

*Dropouts:* To introduce dropouts to the simulated data, a probability is initially assigned to the expression of each gene in each of the simulated cells not being a dropout. This probability is modeled as a logistic function of the expression of the gene in that cell, so that a high expression value is less likely to be zeroed out. Subsequently, this probability is used as the parameter of a Bernoulli distribution from which a binary variable is sampled to indicate whether the gene is not a dropout in the cell.

### 3.4. Datasets

We first simulate "ground-truth" gene expression data as described in §3.2. We then add "technical noise" to the ground truth data, mimicking the nature of measurement errors attributed to scRNA-seq technology, as described in §3.3. For our experiments, we reproduce the steady state simulation datasets described in the SERGIO paper (DS1, DS2, and DS3). The properties of these datasets is given in Table 1. The Gene Regulatory Networks used for the generation of these datasets are a subset of the experimentally validated networks for E. coli and S. cerevisiae.

All three datasets have a clean and a noisy version, where noise is later added to the dataset. For the noisy version of each of the simulated datasets, Dibaeinia et. al. configured SERGIO to introduce technical noise to an extent that matches the published real scRNA-seq dataset on mouse cerebral cortex generated by Illumina HiSeq, 2000 sequencing methodology. The details of the noise iteration process, and comparison between datasets can be found in the SERGIO paper.

One important aspect that needs to be highlighted is the sparsity of the edges in the gene regulatory networks used in the datasets. The prevalence of edges in the graph is less than 0.2%, which makes the network inference essentially a needle-in-haystack problem.

Table 1. Properties of different datasets used for our experiments.

| Dataset | # Cells | # Genes | # Cell Types | # Regulators | # Edges | Species |
|---------|---------|---------|--------------|--------------|---------|---------|
| DS1 | 2700 | 100 | 9 | 10 | 258 | E. coli |
| DS2 | 2700 | 400 | 9 | 37 | 1155 | S. cerevisiae |
| DS3 | 2700 | 1200 | 9 | 127 | 2713 | E. coli |

## 4. Methodology

### 4.1. Experiment Pipeline

The different datasets produced using the SERGIO simulator (DS1, DS2, DS3) are ultimately used as the input for our network inference method to infer a predicted gene regulatory network. The predicted GRNs and the original input GRNs for the datasets are used to compute the AUC-ROC metric, which is the final performance metric for the task. For the clean versions of the datasets, no changes are made and they are directly fed in as input to the network inference algorithm. The resulting AUC-ROC is considered as the ideal performance for different Zero Imputation methods. For the noisy versions of the datasets, we repeat the same experiment to obtain the lower bound on the performance. Then, for different zero-imputation methods, we apply them to the noisy versions of the three datasets, and the outputs are fed as input to the network inference algorithm, and the resulting AUC-ROC values are used as the metric of performance. The pipeline is better understood from Figure 3.

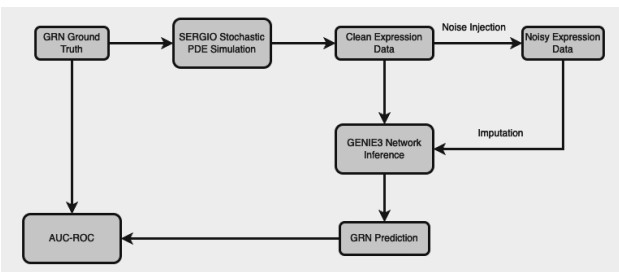

Figure 3. Experiment pipeline flowchart.

### 4.2. Network Inference Algorithm

Similar to the SERGIO paper, we chose to use the GENIE3 algorithm (Huynh-Thu et al., 2010) for network inference. This choice was made to ensure that the data reproduction and network inference pipeline were in-line with the results obtained by Dibaeinia et. al. GENIE3 captures nonlinear relationships between genes and calculates feature importance scores for each gene by employing an ensemble of regression trees. These scores, derived from the strength of gene associations within the dataset, enable the method

to rank potential regulatory links between genes, thereby facilitating the construction of a predicted gene regulatory network (GRN). It should be noted that GENIE3 can take in a list of regulators and incorporate that in the inference pipeline, as was done by (Dibaeinia & Sinha, 2020) when benchmarking the MAGIC (van Dijk et al., 2018) imputation method. We do not provide this regulator set in our experiments.

### 4.3. Evaluated Imputation Methods

For our benchmarking study, we have evaluated the following zero-imputation methods.

*MAGIC:* MAGIC (van Dijk et al., 2018) constructs a graph with cells as nodes and edges representing gene expression similarities, then diffuses information across this graph to impute missing values, effectively smoothing the data while preserving biological variability.

*SAUCIE:* SAUCIE (Amodio et al., 2019) uses a sparse autoencoder to simultaneously perform clustering, imputation, and dimensionality reduction by learning a compressed representation of the data, allowing it to effectively impute missing values, cluster similar cells, and create low-dimensional embeddings that capture key features of gene expression profiles.

*scScope:* ScScope (Deng et al., 2019) uses a neural network to learn a latent representation of the data, capturing complex gene expression patterns to accurately impute missing values, thereby enhancing the quality and resolution of single-cell data for downstream analysis.

*DeepImpute:* DeepImpute (Arisdakessian et al., 2019) uses neural networks to predict and fill missing gene expression values by leveraging observed data and training on subsets of genes to enhance accuracy, scalability, and overall data quality.

*scVI:* scVI (Lopez et al., 2018) uses variational autoencoders (VAEs) to model gene expression distributions. By learning a latent representation, scVI captures biological variability and technical noise. Its Bayesian approach also provides uncertainty estimates for predictions, and is scalable to large datasets.

*kNN-Smoothing:* KNN-Smoothing (Wagner et al., 2018) imputes data by first identifying each cell's k nearest neighbors based on gene expression similarity, followed by averaging the expression values of these neighbors to smooth the data.

*Table 2.* AUC-ROC on network inference task with GENIE3.

| Imputation Method | DS1 | DS2 | DS3 |
|---|---|---|---|
| Clean (Dibaeinia & Sinha, 2020) | $0.685 \pm 0.005$ | $0.806 \pm 0.003$ | $0.825 \pm 0.003$ |
| Noisy (Dibaeinia & Sinha, 2020) | $0.478 \pm 0.003$ | $0.444 \pm 0.003$ | $0.455 \pm 0.003$ |
| MAGIC (van Dijk et al., 2018) | $0.472 \pm 0.006$ | $0.489 \pm 0.002$ | $0.504 \pm 0.003$ |
| SAUCIE (Amodio et al., 2019) | $0.524 \pm 0.022$ | $0.439 \pm 0.016$ | $0.481 \pm 0.013$ |
| scScope (Deng et al., 2019) | $0.491 \pm 0.051$ | $0.464 \pm 0.027$ | $0.478 \pm 0.024$ |
| DeepImpute (Arisdakessian et al., 2019) | $0.530 \pm 0.006$ | $0.502 \pm 0.005$ | $0.411 \pm 0.003$ |
| scVI (Lopez et al., 2018) | $0.492 \pm 0.020$ | $0.505 \pm 0.011$ | $0.500 \pm 0.007$ |
| kNN-Smoothing (Wagner et al., 2018) | $0.513 \pm 0.020$ | $0.496 \pm 0.005$ | $0.480 \pm 0.006$ |

## 5. Results and Implications

The results of our benchmarking experiments are provided in Table 2. We surprisingly observe that none of the tested imputation methods seem to perform well on the new task, with AUC-ROC across all imputation methods close to 0.5. On the three datasets, GENIE3 network inference algorithm works with relatively high AUC-ROC ($> 0.8$) for DS2 and DS3, while the performance on DS1 is significantly lower (0.68). This is likely due to DS1 having a significantly lower number of genes. It should also be noted that since SERGIO uses stochastic differential equations to model the expression matrix, it is unreasonable to assume that any inference method would show perfect performance (AUC-ROC=1). With the addition of noise and dropouts, we see that the performance drops to random, implying that the relevant information was lost for the model to infer the network. With an ideal imputation algorithm, we would be able to impute the noisy dataset to get performance close to clean data. The tested imputation methods do not perform significantly better than the noisy data, signifying that whatever necessary information was present in the clean data for network inference was not restored at all after imputation.

## 6. Discussion

While our work aimed to define a gold standard for determining the quality of zero imputation methods using a synthetic data generation methodology with controllable ground truth, the end-result was far from the desired outcome. We observed that several standard zero imputation methods provided highly variable results on the network inference task on real-world datasets and performed poorly on synthetically generated data from realistic simulation environments. The gap from the defined ground truth is significantly far that we are unable to convincingly determine a quantitative measure that truly can compare the quality of different zero imputation methods; in fact, it raises a larger question on the correctness of zero imputation.

By filling in zeros with estimated values, the true variability and heterogeneity inherent in single-cell data might be masked, potentially skewing downstream analyses like clustering and differential expression. Imputation assumes a certain model or distribution of gene expression, which may not accurately reflect the underlying biological processes. If these assumptions are incorrect, the imputed values might distort the actual expression patterns, leading to misleading conclusions. Moreover, imputation can obscure the natural sparsity of single-cell data, which is a characteristic feature reflecting the stochastic nature of gene expression. Masking this sparsity with imputed values can reduce the data's ability to reveal true biological differences between cell types or states.

We acknowledge that our work may have several limitations in our methodology given the complexity of the underlying biological processes, data generation methodologies and the nuances in the individual algorithmic pipelines. Our benchmark uses the SERGIO simulator, which has a particular set of assumptions and its own limitations. We also note the scarcity of gene-regulation based simulation frameworks that are used for benchmarking scRNA-seq imputation methods as a potential research direction worthy of further thought. We also believe that further work is required on defining mechanisms to standardize pre-processing pipelines including the right way to perform zero imputation for scRNA-seq data.

## 7. Reproducibility

The code for our experiments is available at `https://github.com/ankitbha/dfdl_imputation/`.

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
