# OpenReview forum: "Limitations of scRNA-seq Zero-Imputation Methods for Network Inference"
_ICML.cc/2024/Workshop/ML4LMS — ML4LMS Poster_

### Official Review · Reviewer_Z92n · 2024-05-30
**Concerns about the evaluation of diverse imputation techniques and clarity on noise simulation parameters.**

**Rating:** 6
**Confidence:** 4

**Review:**

In this paper, the authors investigated the limitations of zero-imputation methods in scRNA-seq data, demonstrating that these methods introduce significant variability in gene regulatory network (GRN) inference. Using both real-world and synthetic datasets, the study revealed that existing imputation techniques fail to reliably recover ground truth networks, highlighting the challenges in defining a gold standard for zero-imputation. While evaluating different zero-imputation methods is meaningful, I have two major concerns regarding the introduction and simulation.

1. The authors mentioned statistical methods such as ZINB-WaVE and scImpute, Bayesian frameworks and machine learning algorithms like MAGIC and SAVER in the introduction. However, only the performance of MAGIC was evaluated in the simulations, with other methods not mentioned subsequently. Several imputation methods have been developed recently, which can be categorized based on the data structure considered: a) Impute dropouts by learning a gene-gene similarity matrix and smoothing the data over similar genes, or by encoding gene expression measurements into a low-dimensional space and decoding the imputed values from these representations, such as Deepimpute (Arisdakessian C. et al. (2019) Deepimpute: an accurate, fast, and scalable deep neural network method to impute single-cell RNA-seq data. Genome Biol., 20, 1–14); b) Impute dropout values by learning a cell-cell similarity matrix and smoothing the data over similar cells, or by grouping the cells into clusters and then smoothing the data within clusters, such as scImpute (Li W.V., Li J.J. (2018) An accurate and robust imputation method scimpute for single-cell RNA-seq data. Nat. Commun., 9, 997); c) Impute dropouts using low-rank matrix approximation, such as ALRA (Linderman, G.C., Zhao, J., Roulis, M. et al. (2022) Zero-preserving imputation of single-cell RNA-seq data. Nat Commun., 13, 192). It would be better to provide a comprehensive evaluation.

2. The noise simulation process should be clearly detailed to avoid confusion. The authors mentioned that "technical noise" was added to ground truth data. However, it is unclear what constitutes "technical noise." Does it include outlier genes, library size variations, dropouts, or all of these factors? Additionally, how much noise was added, and what parameters were used? In Table 2, performance drops to random levels with the addition of noise and dropouts, suggesting that the noise may have been excessive, distorting the original distribution. It would be valuable to evaluate the performance of imputation methods across different levels of noise and different factors of noise to determine which zero-imputation method is most robust.

---

### Official Review · Reviewer_CGNc · 2024-06-05
**A good benchmark work for GRN inference**

**Rating:** 7
**Confidence:** 3

**Review:**

In the manuscript titled “Limitations of scRNA-seq Zero-Imputation Methods for Network Inference,” the authors address a novel and compelling question regarding the influence of imputation methods on gene regulatory network (GRN) inference from single-cell RNA sequencing data. The paper is well-structured and easy-to-follow, making contributions to a less explored area. However, I have some recommendations for improvement:

1.	Robustness of GENIE3: It would be beneficial if the authors could provide details on the robustness of the GENIE3 algorithm. Specifically, does it maintain consistent performance across multiple runs and under minor data perturbations?

2.	Imputation Method Consistency: Similarly, the robustness of the imputation method should be evaluated to understand its performance stability across different runs and in response to small perturbations.

3.	Separating Sources of Impact: I recommend conducting experiments to differentiate the effects of data manipulation from the imputation process itself. This could involve systematically introducing noise or dropouts to clean data and observing both the performance of GENIE3 and the ability of imputation methods to reconstruct the original dataset.

4.	Enhanced Metrics for GRN Inference: Adding more comprehensive metrics, such as Recall@k, would provide a clearer picture of the GRN inference quality post-imputation, aiding in the evaluation of the methods used.

5.	Diversity of GRN Topologies: Expanding the study to include a broader array of GRN topologies could enhance the generalizability of the findings. The authors might consider integrating GRN topologies from resources like TrajectoryNet to enrich the analysis.

---

### Official Review · Reviewer_nRhg · 2024-06-12
**The authors proposed to benchmark scRNA-seq imputation methods for GRN inference. They compared the methods on both real and simulated data.**

**Rating:** 7
**Confidence:** 5

**Review:**

## Summary

The authors proposed to benchmark scRNA-seq imputation methods for GRN inference. They compared the methods on both real and simulated data.

## Comments

**Real Data Evaluation**

I believe every method could lead to different results that can be interpreted biologically differently, so that's not that surprising to me. If they can generate very similar results, is there any need for us to build imputation methods? We could just use the raw data. I think the key is which one reflects the biological truth. I don't think such a comparison with real data generated GRN reflects this objective.

**Simulation Data Evaluation**

* One interesting way for comparison is the stability of each method itself, i.e., can they lead to high-consensus or similar results when we alter the input data slightly? This can be achieved by subsampling the input data, repeating the analysis procedure several times, and comparing the results for each method.

* There is a GRN benchmarking paper (https://doi.org/10.1038/s41592-019-0690-6) that the authors might benefit from by checking out what metrics they used.

* Personally, I have concerns about evaluation methods on simulation data as I believe simulation data do not reflect real biology. I worry that there might be many in-silico perturbations or manually added noise, which would deviate the data structure and lead to a shift in the data's properties or characteristics. Thus, none of the methods might be applicable to such synthetic data.